# Serum LH Level on the Day of hCG Administration as a Predictor of the Reproductive Outcomes in Ovulation Induction Cycle Frozen–Thawed Embryo Transfer

**DOI:** 10.3390/jpm13010052

**Published:** 2022-12-27

**Authors:** Qingqing Shi, Yue Jiang, Na Kong, Chenyang Huang, Jingyu Liu, Xiaoyue Shen, Yanxin Sun, Feifei Lu, Jie Mei, Jianjun Zhou

**Affiliations:** 1Center for Reproductive Medicine and Obstetrics and Gynecology, Nanjing Drum Tower Hospital, Nanjing University Medical School, Nanjing 210008, China; 2Center for Molecular Reproductive Medicine, Nanjing University, Nanjing 210008, China

**Keywords:** serum LH, ovulation induction, clinical pregnancy, live birth

## Abstract

**Purpose**: To evaluate the clinical utility of serum luteinizing hormone (LH) level in predicting frozen embryo transfer (FET) outcomes among the patients with an ovulation induction (OI) cycle. **Methods**: A total of 250 patients who underwent OI cycle frozen–thawed embryo transfer from January 2018 to June 2020 in Drum Tower Hospital affiliated with Nanjing University Medical School were retrospectively analyzed. The primary outcomes were clinical pregnancy rate and the live birth rate. **Results**: The results showed that a significant difference in Serum LH level on the day of human chorionic gonadotropin (hCG) administration was observed between the clinical pregnancy group and no clinical pregnancy group (*p* = 0.002), while there was no significant difference between the live birth group and no live birth group (*p* = 0.06). Multiple logistic regression analysis of factors related to clinical pregnancy showed serum LH level on the day of hCG administration was related to improved clinical pregnancy rate (OR was 1.02, 95% CI: 1.0–1.03, *p* = 0.02), while serum LH level had no significant effect on live birth rate. The ROC curves revealed the serum LH level was significantly correlated with clinical pregnancy rate and live birth rate; the cut-off point of serum LH level on the day of hCG administration was 8.46 miu/mL for clinical pregnancy rate (AUC 0.609, *p* = 0.003). **Conclusion**: In patients with OI FET, serum LH level on the day of hCG administration might be a biomarker for the prediction of clinical pregnancy. The prediction that patients who underwent OI cycle frozen–thawed embryo transfer with serum LH levels below 8.46 mIU /mL might be pregnant appeared to be meaningful.

## 1. Introduction

Since the successful pregnancy of the first frozen–thawed embryo transfer (FET) in 1983, the rapid development of vitrification has contributed to the development of FET as a conventional treatment for assisted reproductive technology (ART), which is widely used in pre-implantation genetic diagnosis/screening (PGD/PGS), as well as patients with elevated progesterone, ovarian hyperstimulation syndrome (OHSS), decreased ovarian reserve, and those with the need for preconditioning such as patients with adenomyosis [1]. FET outcomes are affected by multiple factors, including maternal age, ovarian function, embryo quality, endometrial receptivity, and comorbidities [2,3]. Among them, endometrial receptivity is widely recognized as one of the key factors for FET success. The endometrium allows embryo implantation only at specific times and under certain conditions, which is known as the “implantation window”, and this window is usually within 3 to 5 days after fertilization. In the FET cycle, endometrial growth is promoted by endogenous or exogenous estrogen, and later endometrium appears to transform under the action of progesterone and has the ability to accept embryos similar to the implantation window. Previous studies have focused on regulation of the opening and closing of the implantation window by estrogen and progesterone [4,5]. Luteinizing hormone (LH) is a hormone produced by the pituitary gland. The peak before ovulation plays a role in the final maturation and ovulation of oocytes, and is an important basis for clinical decision-making of ART. The production of LH is controlled by the hypothalamic gonadotropin-releasing hormone and regulated by positive and negative feedback of the ovary. The peak of LH release during normal menstrual cycle is closely related to ovulation. First, the serum LH peak appears, and then the ovary begins to ovulate after 24~36 h; thus, the serum LH peak can be monitored during the menstrual cycle to determine the best time for conception. The purpose of this study was to evaluate the clinical value of serum luteinizing hormone in predicting the outcome of FET in patients with ovulation induction (OI) cycle from the point of view of clinical pregnancy and live birth.

## 2. Materials and Methods

### 2.1. Patient Population

This study included 250 patients who underwent OI cycle frozen–thawed embryo transfer (OI-FET) from January 2018 to June 2020 at Drum Tower Hospital affiliated with Nanjing University School of Medicine. All patients with at least a good cleavage embryo quality met passed the inclusion criteria. Patients with known endocrine disorders and uterine anomalies were excluded. The institutional ethics committee approved the study, and all of the patients provided informed consent.

### 2.2. Serum Hormonal and Ultrasound Assays

On the third day of the menstrual cycle, serum follicle-stimulating hormone (FSH), estradiol (E_2_), and LH were assessed in peripheral blood, and on the same day, antral follicle count (AFC) was assessed with transvaginal ultrasonography with follicles larger than 4mm in diameter in both ovaries. On the day of human chorionic gonadotropin (hCG) administration, we collected peripheral blood samples to assess serum E_2_ and LH levels, and endometrial thickness was examined by transvaginal ultrasonography.

### 2.3. Treatment Protocol

Ovarian stimulation was oral tamoxifen (TMXF, Jiangsu Hengrui Medicine) 20 mg/d started on day 3 of menstrual cycle and continuing for 5 consecutive days, and human menopausal urinary gonadotropin (HMG, Livzon Group Livzon Pharmaceutical) was intramuscularly injected later or intramuscular injection of HMG on the third day of menstrual cycle. The starting dose of HMG was 75 IU/d. According to the ovarian response, doses of HMG would be further adjusted. When the mean diameter of at least one follicle reached 18 mm, intramuscular injection of 5000 IU to 10,000 IU of human chorionic gonadotropin (hCG) was administered to trigger ovulation. Cleavage embryos were transferred after 5 days of hCG administration. A quantity of 40 mg dydrogesterone was supported daily in the luteal phase and continued until 10 weeks of pregnancy.

Clinical pregnancy was defined as transvaginal ultrasonography showing a gestation with fetal heart activity after embryo transfer of 4 to 5 weeks, and clinical pregnancy rate as the number of clinical pregnancy cycles divided by the total number of transfer cycles. Live birth rate was defined as the delivery of a healthy infant after 28 weeks of gestation, and live birth rate as the number of live birth cycles divided by the total number of transfer cycles.

### 2.4. Statistical Analysis

For the continuous variable, tests were used to compare differences between pregnancy and no pregnancy cycles as well as live birth and no live birth cycles, such as body mass index, duration of infertility, endometrial thickness, and number of transferred embryos. The ability of selected variables to predict the CPR and LBR was analyzed and evaluated using a receiver operating characteristic (ROC) curve. Optimal cut-off level was determined by the combination of sensitivity and specificity closest to the optimal. All the results of the logistic regression models were expressed as odds ratios (ORs) and 95% confidence intervals (Cis). A = 0.05 was considered statistically significant, and all analytic probabilities (*p*-value) were two-sided.

## 3. Results

The basic demographic characteristics of the participants are presented in Table 1. A total of 250 patients were retrospectively analyzed; of the 250 patients, 124 achieved clinical pregnancy (49.6%) and 87 delivered (35.2%).

In Table 2, comparison between the clinical pregnancy group (*n* = 124) and non-pregnancy group (*n* = 126) showed that the women’s mean age and serum LH levels on the hCG day in pregnant group were lower than those in the non-pregnant group (age: 32.11 ± 4.39 vs. 36.44 ± 5.97; serum LH: 14.32 ± 15.88 vs 23.42 ± 28.51. *p* < 0.01). The mean AFC, endometrial thickness, total embryo number, and number of embryos transferred were higher in the pregnancy group than in the non-pregnant group (AFC: 15.12 ± 6.48, 12.54 ± 6.21. *p* < 0.01; endometrial thickness: 8.69 ± 1.83, 8.11 ± 1.23, *p* < 0.01; total embryos: 1.85 ± 0.35, 1.63 ± 0.49, *p* < 0.01, the number of embryos transferred: 6.02 ± 3.67, 4.19 ± 3.02, *p* < 0.01). There was no difference in serum LH on the hCG day between the live birth group (*n* = 87) and the non-live birth group (*n* = 163) (serum LH: 15.42 ± 16.6 vs. 20.45 ± 26.61. *p* = 0.06). Additionally, there was also no difference in other variables such as basic biochemical results, body mass index, duration of infertility, total dose of HMG, and number of IVF cases between the two groups.

Furthermore, we used an ROC curve to evaluate the predictive values of selected variables for clinical pregnancy and live birth (Figure 1).

As shown in Table 3, multivariate regression analysis used to predict clinical pregnancy showed that serum LH (AUC 0.609, *p* = 0.003, cut-off value = 8.46 miu/mL), age (AUC 0.707, *p* < 0.01. cut-off value = 3 7.5), AFC (AUC 0.619, *p* < 0.01, cut-off value = 10.5), endometrial thickness (AUC 0.583, *p* = 0.02, cut-off value = 7.75 mm), the total embryo number (AUC 0.67, *p* < 0.01, cut-off value = 4.5), and the average number of embryos transferred (AUC = 0.614, *p* = 0.02, cut-off value = 1.5) had accuracy by the area under the curve.

As shown in Table 4, the area under the ROC curve was used to analyze the accuracy of serum LH (AUC 0.55, *p* = 0.038, cut-off value = 8.46 miu/mL), age (AUC 0.679, *p* < 0.01, cut-off value = 33.5), the total embryo number (AUC 0.64, *p* < 0.01, cut-off value = 4.5), endometrial thickness (AUC 0.623, *p* = 0.001, cut-off value = 8.75 mm), and the average number of embryos transferred (AUC = 0.594, *p* = 0.015, cut-off value = 1.5) in predicting live birth.

In Table 5, with multivariate regression analysis, serum LH (OR 1.20, *p* = 0.02), age (OR 1.13, *p* < 0.01), total embryos (OR 0.86, *p* = 0.04), and the number of embryos transferred (OR 0.38, *p* < 0.01) were significantly correlated with clinical pregnancy, whereas age (OR 0.88, *p* < 0.01), AFC (OR 0.92, *p* < 0.01), total embryos (OR 1.18, *p* < 0.01), endometrial thickness (OR 1.32, *p* < 0.01), and the number of embryos transferred (OR 2.82, *p* < 0.01) were significantly correlated with live birth.

## 4. Discussion

The rapid development of FET technology has created a new era of ART. Although FET cycles have the advantages of an increased cumulative pregnancy rate, low cost, and simple treatment, a relatively low pregnancy rate is one of the current problems. Synchronization between frozen–thawed embryos and endometrial development is an important factor in embryo implantation. Successful embryo implantation requires a well-acceptable endometrium. Studies have shown that two-thirds of pregnancy failures are caused by defects in endometrial receptivity. Therefore, the establishment of endometrial receptivity is the key to embryo implantation and achievement of a normal pregnancy [6]. Simultaneously, accurate selection of the timing of transplantation of frozen–thawed embryos is essential to ensuring the clinical pregnancy rate of frozen–thawed embryos. Clinically, the embryo transfer date is usually determined according to ovulation and hormone levels. The window from 6 to 10 days post-ovulation, which is equivalent to 5-8 days post-LH peak, is the most suitable period for endometrial implantation and is also known as the implantation window, and a peak period of progesterone secretion occurs in the middle of the luteal phase and generally lasts about 48 h [7,8]. At the genetic level, it has been found that chromosomes 4, 9, and 14 are closely related to embryo implantation. During the implantation window, there was a total of 1543 gene expression changes, including genes that regulate apoptosis, immunity, transcription, metabolism [9].

Endometrial preparation methods in the frozen–thawed cycle mainly include natural cycle/modified natural cycle FET comparisons, HRT FET versus HRT plus GnRHa suppression and the OI cycle. Many studies suggest that there is no evidence in the FET endometrial preparation program that any of the programs have the most advantages.

A number of studies suggest that no evidence is available to support that any programs have the most advantages among FET endometrium preparation programs [10,11,12,13]. Clinically, the most appropriate program is selected according to the patient’s condition. For example, for patients with a thin endometrium, irregular menstruation, or ovulation disorders, the OI cycle program, a scheme for achieving endometrium preparation by inducing follicular development through a mild OI protocol, is usually most suitable. On the one hand, ovarian stimulation can produce endogenous LH and a near-natural endometrium, which can produce the natural estrogen and progesterone required during the luteal phase and early pregnancy. On the other hand, injection of hCG trigger is an important part of obtaining an ideal OI. In general, the timing and dose of hCG are determined mainly by the size/number of follicle diameters and hormone levels in peripheral blood. The OI cycle program aims to induce superior follicles to mature further and controls ovulation timing. The LH level should be routinely tested on the day of HCG injection to determine the hCG injection time. Previous data indicate high levels of endometrial LH/hCG receptor expression and LH secretion during the “implantation window”, which is conducive to embryo implantation. However, premature elevation of LH is detrimental to embryo implantation and may lead to early abortion [14,15]. LH is currently generally accepted to affect the intima indirectly through the action of progesterone. During the implantation window, LH directly binds to the LH receptors on the corpus luteum and endometrium, providing support for the development of the corpus luteum and endometrium, and improving endometrial receptivity [16,17]. Low concentrations of LH indirectly affect the expression of E2 and progesterone receptors in the endometrium, reducing the rate of early embryo implantation. Proper addition of active LH can enhance the expression of the corresponding factors on the endometrium, thereby improving endometrial receptivity and significantly increasing the implantation rate and pregnancy rate of the thawed embryo [18]. This study was conducted to investigate the predictive value of endogenous LH peaks for clinical outcomes in patients with OI-FET during the OI cycle, and the results suggest excessive serum LH levels on the hCG injection day can reduce clinical pregnancy rates. In the natural cycle, the endometrium cells and granulosa cells of the preovulatory follicle secrete E2 under the synergistic action of FSH and LH, and the dominant follicles mature to form the E2 peak, which has a positive feedback effect on the hypothalamic LH-RH and produces an LH peak to induce ovulation. After luteal formation and secretion of progesterone and E2 by luteal cells, LH activates 17-hydroxylase on the ovarian membrane cells, causing progesterone to decrease, and in the late follicles, it can act on progesterone receptors of granulosa cells, causing progesterone to rise [19,20]. We hypothesize that an abnormal increase in LH levels in the late follicular phase during the freezing cycle triggers an increase in progesterone levels, thereby reducing endometrial receptivity and ultimately altering clinical pregnancy outcomes.

Under normal physiological conditions, blood LH is directly secreted and released into the blood by the pituitary gland under the control of hypothalamic gonadotropin-releasing hormone (GnRH), which directly reflects the secretion of the maternal body. The peak value of LH level before ovulation plays an essential role in the final maturation and ovulation of oocytes. The appropriate serum LH concentration in assisted pregnancy is an important condition to ensure the normal development and maturation of follicles. The level of LH is directly related to endometrial receptivity and the pregnancy outcome of the OI cycle. Therefore, the monitoring LH levels can determine the timing of embryo transfer and is an important basis for clinical decision-making in ART.

The advantage of our study is that it is the first known study of the relationship between serum LH and clinical pregnancy rate in the ovulation induction cycle in patients with frozen–thawed embryo transfer. In this study, the predictive value of LH was similar for age and endometrial thickness, and the mean number of embryos transferred were recognized as predictive indicators [21,22,23]. After controlling for other independent variables (age, AMH, basal FSH, the number of retrieved oocytes), the significant correlation between serum LH and clinical pregnancy was still very strong (OR was 1.02, 95% CI: 1.0–1.03, *p* = 0.02).

## 5. Conclusions

In conclusion, our study demonstrates that OI-FET has a certain effect on pregnancy outcomes besides AFC, age, endometrial thickness, number of transplanted embryos, number of embryos reported in previous studies [24]. Patients younger than 37.5 years old, with more than 1.5 transplanted embryos, 4.5 quality embryos, and LH levels lower than 8.46 mIU/mL on hCG trigger day have high clinical pregnancy rates and good clinical outcomes. Therefore, the LH level on the hCG trigger day can be considered as a reference indicator for predicting the outcome of FET and guiding clinical decision-making.

## Figures and Tables

**Figure 1 jpm-13-00052-f001:**
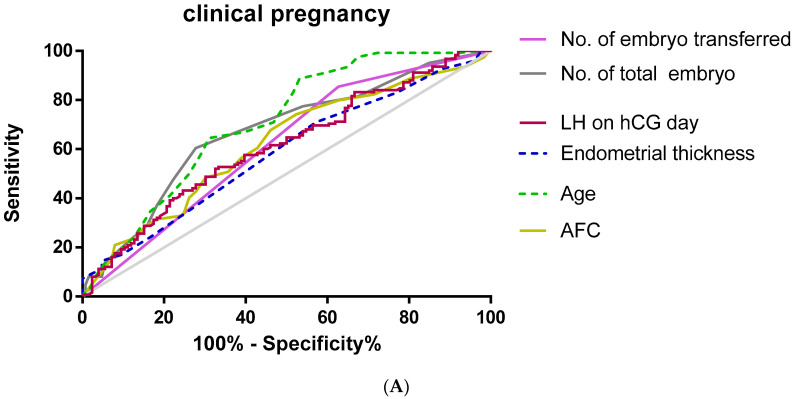
Receiver operating characteristic (ROC) curve for age, AFC, LH on hCG day, endometrial thickness, no. of embryos transferred, and no. Of total embryos for prediction of clinical pregnancy and live birth. (**A**) Clinical pregnancy; (**B**) live birth.

**Table 1 jpm-13-00052-t001:** Demographics, patient characteristics, and clinical outcomes.

Clinical Parameters	Frozen-Thawed Enbryo Transfer (*n* = 250)
Age of woman (yr)	34.29 ± 5.66
LH on hCG day (mIU/mL)	18.9 ± 23.86
Endometrial thickness (mm)	8.4 ± 1.58
BMI (kg/m^2^)	22.8 ± 2.90
AFC (#)	13.8 ± 6.47
Basal FSH (MIU/mL)	7.98 ± 3.23
Basal LH (MIU/mL)	4.87 ± 2.95
Basal E2 (pg/nL)	50.33 ± 32.67
Duration of infertility(yr)	4.37 ± 3.62
E2 on hCG day (pg/mL)	717.31 ± 549.55
No. of embryos transferred	1.74 ± 0.44
Tine of HMG(d)	9.41 ± 5.52
No. of total retrieved oocytes	11 ± 6.63
Total HMG dose (IU)	941.56 ± 879.44
No. of total retrieved enbryos	5.1 ± 3.47
No. of ICSI cases	25.2% (63/250)
Clinical pregnancy rate	49.6% (124/250)
Live birth rate	35.2% (88/250)
Primary infertility	40.4% (101/250)
Secondary infertility	59. 61 (149/250)

Values are shown as mean ± SD.

**Table 2 jpm-13-00052-t002:** Patient and cycle characteristics by pregnancy outcome.

	Pregnant Group (*n* = 124)	Non—Pregnant Group (*n* = 126)	*p*-Value	Live Birth (*n* = 87)	Nonlive Birth (*n* = 163)	*p*-Value
Age of woman (yr)	32.11 ± 4.39	36.44 ± 5.97	0	31.86 ± 4.26	35.59 ± 5.9	0
LH on hCG day (mIU/mL)	14.32 ± 15.88	23.42 ± 28.51	0.002	15.42 ± 16.6	20.45 ± 26.61	NS
Endometrial thickness (mm)	8.69 ± 1.83	8.11 ± 1.23	0.003	8.84 ± 1.77	8.17 ± 1.43	0.003
BMI (kg/m^2^)	22.8 ± 3.1	22.81 ± 2.71	NS	22.47 ± 2.94	22.95 ± 2.85	NS
AFC (#)	15.12 ± 6.48	12.54 ± 6.21	0.001	14.51 ± 6.33	13.45 ± 6.55	NS
Basal FSH (mIU/mL)	7.5 ± 2.99	8.45 ± 3.4	NS	7.69 ± 3.33	8.13 ± 3.17	NS
Basal LH (mIU/mL)	5.09 ± 3.09	4.64 ± 2.79	NS	4.91 ± 2.93	4.82 ± 2.96	NS
Basal E2 (pg/mL)	50.26 ± 30.7	50.39 ± 34.62	NS	50.18 ± 28.79	50.58 ± 34.68	NS
Duration of infertility(yr)	3.93 ± 2.65	4.8 ± 4.34	NS	4.16 ± 2.76	4.48 ± 4.0	NS
E2 on hCG day (pg/mL)	690.85 ± 588.45	717.31 ± 509.02	NS	640.3 ± 441.32	737.96 ± 597.87	NS
No. of enbryos transferred	1.85 ± 0.35	1.63 ± 0.49	0	1.86 ± 0.34	1.67 ± 0.46	0
Time of HMG(d)	9.71 ± 5.28	9.06 ± 5.8	NS	9.11 ± 4.35	9.53 ± 6.13	NS
Total HMG dose (IU)	995.8 ± 887.56	877.42 ± 870.12	NS	902.96 ± 626.02	957.13 ± 950.26	NS
No. of total embryos	6.02 ± 3.67	4.19 ± 3.02	0	6.18 ± 3.91	4.51 ± 3.06	0.001
No. of IVF cases	75.81% (94/124)	73.81% (93/126)	NS	78.16% (68/87)	73% (119/163)	NS
Primary infertility	42.74% (53/124)	38.1% (48/126)	NS	45.97 (40/87)	37.42% (61/163)	NS

Values are shown as mean ± SD. Differences between means were tested by *t*-test for equality of means and differences in rates were tested by chi-square test. NS = not significant.

**Table 3 jpm-13-00052-t003:** ROC analysis of the area under the curve and the specific cut-off value for predicting clinical pregnancy.

	Area under Curve	95% CI	Cut-Off Value	Sensitivity	Specificity	*p*-Value
LH on hCG day (mIU/mL)	0.609	0.539–0.678	8.46	51.6	67.5	0.003
Age of woman (yr)	0.707	0.642–0.771	37.5	88.7	46.8	0
Endometrial thickness (m)	0.583	0.513–0.654	7.75	71	42.9	0.023
AFC (#)	0.619	0.549–0.689	10.5	74.2	47.6	0.001
No. of enbryos transferred	0.614	0.544–0.684	1.5	85.5	37.3	0.002
No. of total enbryos	0.67	0.603–0.737	4.5	60.5	72.7	0

**Table 4 jpm-13-00052-t004:** Area under curve of ROC analysis and specific cut-off values for prediction of live birth.

	Area under Curve	95% CI	Cut-Off Value	Sensitivity	Specificity	*p*-Value
LH on hCG day (mIU/mL)	0.55	0.476–0.625	8.46	49.4	62	0.038
Age of woman (yr)	0.679	0.514–0.745	33.5	67.8	63.2	0
Endometrial thickness (m)	0.623	0.549–0.698	8.75	47.1	73	0.001
AFC (#)	0.552	0.478–0.626	10.5	71.3	41.1	NS
No. of enbryos transferred	0.594	0.522–0.665	1.5	86.2	32.5	0.015
No. of total enbryos	0.64	0.569–0.711	4.5	60.9	65	0

**Table 5 jpm-13-00052-t005:** Multivariate logistic regression analysis of factors related to clinical pregnancy and live birth.

	Clinical Pregnancy			Live Birth		
	*p*	OR	95% CI	*p*	OR	95% CI
LH on hCG day (mIU/mL)	0.02	1.02	1.0–1.03	0.241	0.992	0.978–1.006
Age of woman (yr)	0	1.13	1.07–1.2	0	0.884	0.830–0.942
Endometrial thickness (mm)	0.108	0.85	0.69–1.04	0.008	1.317	1.075–1.612
AFC (#)	0.3	1.03	0.98–1.09	0.005	0.92	0.867–0.975
No. of enbryos transferred	0.006	0.38	0.19–0.75	0.008	2.816	1.308–6.061
No. of total enbryos	0.04	0.86	0.78–0.95	0.001	1.183	1.072–1.306

## Data Availability

The datasets used and/or analyzed during the current study are available from the corresponding author upon reasonable request.

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
