# Peer review of "Serum LH Level on the Day of hCG Administration as a Predictor of the Reproductive Outcomes in Ovulation Induction Cycle Frozen–Thawed Embryo Transfer"

_jpm, 2022, doi:10.3390/jpm13010052_

Round 1

Reviewer 1 Report

My remarks:

·         You say :«two-thirds of pregnancy failures are caused by defects in endometrial receptivity« Source?

·         If I understand Patient population correctly, your patients had ovulatory menstrual cycles. Why did you stimulate them for FET?

·         What are the advantages of tamoxifen? Why choose it over other protocols?

·         Was estradiol concentration also taken into account (together with follicular diameter) when hCG administration timing was determined?

·         Why didn't you measure preovulatory progesterone concentration?

·         What was the rationale behind AFC assessment in the FET cycle?

·         Your clinical pregnancy rate is very high. Did you perform PGD-A?

·         According to your results did you include LH cut-off in clinical decision-making?

·         A lot of editing is necessary and some corrections

My remarks:

·         You say :«two-thirds of pregnancy failures are caused by defects in endometrial receptivity« Source?

·         If I understand Patient population correctly, your patients had ovulatory menstrual cycles. Why did you stimulate them for FET?

·         What are the advantages of tamoxifen? Why choose it over other protocols?

·         Was estradiol concentration also taken into account (together with follicular diameter) when hCG administration timing was determined?

·         Why didn't you measure preovulatory progesteron concentration?

·         What was the rationale behind AFC assessment in the FET cycle?

·         Your clinical pregnancy rate is very high. Did you perform PGD-A?

·         According to your results did you include LH cut-off in clinical decision-making?

·         A lot of editing is necessary and some corrections

My remarks:

·         You say :«two-thirds of pregnancy failures are caused by defects in endometrial receptivity« Source?

·         If I understand Patient population correctly, your patients had ovulatory menstrual cycles. Why did you stimulate them for FET?

·         What are the advantages of tamoxifen? Why choose it over other protocols?

·         Was estradiol concentration also taken into account (together with follicular diameter) when hCG administration timing was determined?

·         Why didn't you measure preovulatory progesteron concentration?

·         What was the rationale behind AFC assessment in the FET cycle?

·         Your clinical pregnancy rate is very high. Did you perform PGD-A?

·         According to your results did you include LH cut-off in clinical decision-making?

·         A lot of editing is necessary and some corrections

My remarks:

·         You say :«two-thirds of pregnancy failures are caused by defects in endometrial receptivity« Source?

·         If I understand Patient population correctly, your patients had ovulatory menstrual cycles. Why did you stimulate them for FET?

·         What are the advantages of tamoxifen? Why choose it over other protocols?

·         Was estradiol concentration also taken into account (together with follicular diameter) when hCG administration timing was determined?

·         Why didn't you measure preovulatory progesteron concentration?

·         What was the rationale behind AFC assessment in the FET cycle?

·         Your clinical pregnancy rate is very high. Did you perform PGD-A?

·         According to your results did you include LH cut-off in clinical decision-making?

·         A lot of editing is necessary and some corrections

My remarks:

·         You say :«two-thirds of pregnancy failures are caused by defects in endometrial receptivity« Source?

·         If I understand Patient population correctly, your patients had ovulatory menstrual cycles. Why did you stimulate them for FET?

·         What are the advantages of tamoxifen? Why choose it over other protocols?

·         Was estradiol concentration also taken into account (together with follicular diameter) when hCG administration timing was determined?

·         Why didn't you measure preovulatory progesteron concentration?

·         What was the rationale behind AFC assessment in the FET cycle?

·         Your clinical pregnancy rate is very high. Did you perform PGD-A?

·         According to your results did you include LH cut-off in clinical decision-making?

·         A lot of editing is necessary and some corrections

My remarks:

·         You say :«two-thirds of pregnancy failures are caused by defects in endometrial receptivity« Source?

·         If I understand Patient population correctly, your patients had ovulatory menstrual cycles. Why did you stimulate them for FET?

·         What are the advantages of tamoxifen? Why choose it over other protocols?

·         Was estradiol concentration also taken into account (together with follicular diameter) when hCG administration timing was determined?

·         Why didn't you measure preovulatory progesteron concentration?

·         What was the rationale behind AFC assessment in the FET cycle?

·         Your clinical pregnancy rate is very high. Did you perform PGD-A?

·         According to your results did you include LH cut-off in clinical decision-making?

·         A lot of editing is necessary and some corrections

My remarks:

·         You say :«two-thirds of pregnancy failures are caused by defects in endometrial receptivity« Source?

·         If I understand Patient population correctly, your patients had ovulatory menstrual cycles. Why did you stimulate them for FET?

·         What are the advantages of tamoxifen? Why choose it over other protocols?

·         Was estradiol concentration also taken into account (together with follicular diameter) when hCG administration timing was determined?

·         Why didn't you measure preovulatory progesteron concentration?

·         What was the rationale behind AFC assessment in the FET cycle?

·         Your clinical pregnancy rate is very high. Did you perform PGD-A?

·         According to your results did you include LH cut-off in clinical decision-making?

·         A lot of editing is necessary and some corrections

My remarks:

·         You say :«two-thirds of pregnancy failures are caused by defects in endometrial receptivity« Source?

·         If I understand Patient population correctly, your patients had ovulatory menstrual cycles. Why did you stimulate them for FET?

·         What are the advantages of tamoxifen? Why choose it over other protocols?

·         Was estradiol concentration also taken into account (together with follicular diameter) when hCG administration timing was determined?

·         Why didn't you measure preovulatory progesteron concentration?

·         What was the rationale behind AFC assessment in the FET cycle?

·         Your clinical pregnancy rate is very high. Did you perform PGD-A?

·         According to your results did you include LH cut-off in clinical decision-making?

·         A lot of editing is necessary and some corrections

My remarks:

·         You say :«two-thirds of pregnancy failures are caused by defects in endometrial receptivity« Source?

·         If I understand Patient population correctly, your patients had ovulatory menstrual cycles. Why did you stimulate them for FET?

·         What are the advantages of tamoxifen? Why choose it over other protocols?

·         Was estradiol concentration also taken into account (together with follicular diameter) when hCG administration timing was determined?

·         Why didn't you measure preovulatory progesteron concentration?

·         What was the rationale behind AFC assessment in the FET cycle?

·         Your clinical pregnancy rate is very high. Did you perform PGD-A?

·         According to your results did you include LH cut-off in clinical decision-making?

·         A lot of editing is necessary and some corrections

My remarks:

·         You say :«two-thirds of pregnancy failures are caused by defects in endometrial receptivity« Source?

·         If I understand Patient population correctly, your patients had ovulatory menstrual cycles. Why did you stimulate them for FET?

·         What are the advantages of tamoxifen? Why choose it over other protocols?

·         Was estradiol concentration also taken into account (together with follicular diameter) when hCG administration timing was determined?

·         Why didn't you measure preovulatory progesteron concentration?

·         What was the rationale behind AFC assessment in the FET cycle?

·         Your clinical pregnancy rate is very high. Did you perform PGD-A?

·         According to your results did you include LH cut-off in clinical decision-making?

·         A lot of editing is necessary and some corrections

My remarks:

·         You say :«two-thirds of pregnancy failures are caused by defects in endometrial receptivity« Source?

·         If I understand Patient population correctly, your patients had ovulatory menstrual cycles. Why did you stimulate them for FET?

·         What are the advantages of tamoxifen? Why choose it over other protocols?

·         Was estradiol concentration also taken into account (together with follicular diameter) when hCG administration timing was determined?

·         Why didn't you measure preovulatory progesteron concentration?

·         What was the rationale behind AFC assessment in the FET cycle?

·         Your clinical pregnancy rate is very high. Did you perform PGD-A?

·         According to your results did you include LH cut-off in clinical decision-making?

·         A lot of editing is necessary and some corrections

My remarks:

·         You say :«two-thirds of pregnancy failures are caused by defects in endometrial receptivity« Source?

·         If I understand Patient population correctly, your patients had ovulatory menstrual cycles. Why did you stimulate them for FET?

·         What are the advantages of tamoxifen? Why choose it over other protocols?

·         Was estradiol concentration also taken into account (together with follicular diameter) when hCG administration timing was determined?

·         Why didn't you measure preovulatory progesteron concentration?

·         What was the rationale behind AFC assessment in the FET cycle?

·         Your clinical pregnancy rate is very high. Did you perform PGD-A?

·         According to your results did you include LH cut-off in clinical decision-making?

·         A lot of editing is necessary and some corrections

My remarks:

·         You say :«two-thirds of pregnancy failures are caused by defects in endometrial receptivity« Source?

·         If I understand Patient population correctly, your patients had ovulatory menstrual cycles. Why did you stimulate them for FET?

·         What are the advantages of tamoxifen? Why choose it over other protocols?

·         Was estradiol concentration also taken into account (together with follicular diameter) when hCG administration timing was determined?

·         Why didn't you measure preovulatory progesteron concentration?

·         What was the rationale behind AFC assessment in the FET cycle?

·         Your clinical pregnancy rate is very high. Did you perform PGD-A?

·         According to your results did you include LH cut-off in clinical decision-making?

·         A lot of editing is necessary and some corrections

My remarks:

·         You say :«two-thirds of pregnancy failures are caused by defects in endometrial receptivity« Source?

·         If I understand Patient population correctly, your patients had ovulatory menstrual cycles. Why did you stimulate them for FET?

·         What are the advantages of tamoxifen? Why choose it over other protocols?

·         Was estradiol concentration also taken into account (together with follicular diameter) when hCG administration timing was determined?

·         Why didn't you measure preovulatory progesteron concentration?

·         What was the rationale behind AFC assessment in the FET cycle?

·         Your clinical pregnancy rate is very high. Did you perform PGD-A?

·         According to your results did you include LH cut-off in clinical decision-making?

·         A lot of editing is necessary and some corrections

My remarks:

·         You say :«two-thirds of pregnancy failures are caused by defects in endometrial receptivity« Source?

·         If I understand Patient population correctly, your patients had ovulatory menstrual cycles. Why did you stimulate them for FET?

·         What are the advantages of tamoxifen? Why choose it over other protocols?

·         Was estradiol concentration also taken into account (together with follicular diameter) when hCG administration timing was determined?

·         Why didn't you measure preovulatory progesteron concentration?

·         What was the rationale behind AFC assessment in the FET cycle?

·         Your clinical pregnancy rate is very high. Did you perform PGD-A?

·         According to your results did you include LH cut-off in clinical decision-making?

·         A lot of editing is necessary and some corrections

My remarks:

·         You say :«two-thirds of pregnancy failures are caused by defects in endometrial receptivity« Source?

·         If I understand Patient population correctly, your patients had ovulatory menstrual cycles. Why did you stimulate them for FET?

·         What are the advantages of tamoxifen? Why choose it over other protocols?

·         Was estradiol concentration also taken into account (together with follicular diameter) when hCG administration timing was determined?

·         Why didn't you measure preovulatory progesteron concentration?

·         What was the rationale behind AFC assessment in the FET cycle?

·         Your clinical pregnancy rate is very high. Did you perform PGD-A?

·         According to your results did you include LH cut-off in clinical decision-making?

·         A lot of editing is necessary and some corrections

My remarks:

·         You say :«two-thirds of pregnancy failures are caused by defects in endometrial receptivity« Source?

·         If I understand Patient population correctly, your patients had ovulatory menstrual cycles. Why did you stimulate them for FET?

·         What are the advantages of tamoxifen? Why choose it over other protocols?

·         Was estradiol concentration also taken into account (together with follicular diameter) when hCG administration timing was determined?

·         Why didn't you measure preovulatory progesteron concentration?

·         What was the rationale behind AFC assessment in the FET cycle?

·         Your clinical pregnancy rate is very high. Did you perform PGD-A?

·         According to your results did you include LH cut-off in clinical decision-making?

·         A lot of editing is necessary and some corrections

Author Response

Point 1: You say :«two-thirds of pregnancy failures are caused by defects in endometrial receptivity« Source?

Response 1:  Haouzi D, Dechaud H, Assou S, De Vos J, Hamamah S. Insights into human endometrial receptivity from transcriptomic and proteomic data. Reprod
BioMed Online. 2012;24(1):23–34. https://doi.org/10.1016/j.rbmo.2011.09.009.

Achache H, Revel A. Endometrial receptivity markers, the journey to
successful embryo implantation. Hum Reprod Update. 2006;12(6):731–46.
https://doi.org/10.1093/humupd/dml004

Point 2: If I understand Patient population correctly, your patients had ovulatory menstrual cycles. Why did you stimulate them for FET?

Response 2:  In this study, patients had no natural ovulation or thin endometrium, so we stimulate them for FET.

Point 3: What are the advantages of tamoxifen? Why choose it over other protocols?

Response 3:  According to literature reports and our clinical experience, tamoxifen can improve endometrial thickness and pregnancy rate in patients with thin endometrium.

Point 4:  Was estradiol concentration also taken into account (together with follicular diameter) when hCG administration timing was determined?

Response 4:  HCG administration when the dominant follicle diameter is greater than 18mm and estrogen levels are greater than 200pg/ml.

Point 5: Why didn't you measure preovulatory progesterone concentration?

Response 5:  The negative effects of high progesterone on embryo implantation are well recognized, so when progesterone is greater than 1ng/ml, we will abandon embryo transfer considering the decrease of endometrial receptivity. The progesterone of the study population was in the normal range (less than 1ng/ml).

Point 6:    What was the rationale behind AFC assessment in the FET cycle?

Response 6:  AFC examination is an explanation of ovarian function. After all, the pregnancy rate of patients with low ovarian function is low regardless of ET cycle or FET cycle. In order to explain the independent influence of LH, we need to exclude other factors affecting pregnancy.

Point 7:   Your clinical pregnancy rate is very high. Did you perform PGD-A?

Response 7:  The study population did not undergo PGT-A. Our center has been established for more than 20 years, and the clinical and laboratory technologies are in the forefront of the country. The pregnancy rate has been maintained at the leading level in China all year round. But we're hoping for a breakthrough in pregnancy rates.

Point 8:   According to your results did you include LH cut-off in clinical decision-making?

Response 8:  Yes, we will include LH cut-off in clinical decision-making since it is supported by clinical data.

Point 9:     A lot of editing is necessary and some corrections

Response 9:   English language and style have been checked

Reviewer 2 Report

The manuscript “Serum LH level on the day of hCG administration as a predictor of the reproductive outcomes in ovulation induction cycle frozen-thawed embryo transfer” suggests a potential predictor for a patient with female infertility. Differing from previous reports, the authors conducted different approaches to LH values and suggest the possibility of a diagnosis method. Thus, the authors need to describe the possibility of this diagnostic method in practice. In my opinion, I expect how much this method can contribute to infertility treatment.

Some points have to be corrected.

Major points

1. What percentage of patients in the non-pregnant group were lower than an LH value of 8.46 mIU/mL? If it is not o%, it is better to discuss the reason why they have not been pregnant.

2. Similar to this study, are there any reports focusing on the LH value? Are there differences in LH levels among races? Because there may be factors involved with the LH value, for example, lifestyles, such as nutrients, smoking, and alcohol.

3. The LH value has a wide reference range, and there are large individual differences. What kind of approach should be taken to increase the pregnancy rate? Unfortunately, there is no difference in the birth rate even if the pregnancy rate increases.

Minor points

There are no line numbers in the author’s manuscript.

In abstract,

level were significantly correlated with clinical pregnancy rate and live birth rate, the cut-off points of Serum LH level on the day of hCG administration was 8.46 miu/mL for clinical pregnancy rate (AUC 0.609, P = 0.003). Conclusion: In patients with OI FET, Serum LH level on the day of hCG administration might be an biomarker for the prediction of clinical pregnancy. The prediction that

1. Please amend “were” to “was”.

2. Please amend “an” to “a”

In Introduction,

3. Page 1,

as the "implantation window", and this window usually within 3 to 5 days after fertiliza

Please add “is” before “usually”.

4. Page 2,

duction of LH is controlled by the hypothalamic gonadotropin- releasing hormone, and regulated by positive and negative feedback of the ovary. The peak of LH release during normal menstrual cycle is closely related to ovulation ovulation. First, the serum LH peak appears, and then the ovary begins to ovulate after 24~36 h; thus, serum LH peak can be monitored during the menstrual cycle to determine the best time for conception. This study was designed to evaluate the clinical utility of Serum LH to predict the FET out- comes, in terms of clinical pregnancy and live birth, in patients who with ovulation induc- tion(OI) cycle.

There is a space before releasing in gonadotropin-releasing.

There is double ovulation.

I think the sentence needs not “who”.

There is not a space between “(OI)”.

In 2.3. Treatment protocol,

menstrual cycle. The starting dose of HMG were 75 IU /d. According to the ovarian re- sponse ,doses of HMG would further adjusted. When at least one follicle reached a mean diameter larger than 18 mm, intramuscular injection of 5000IU to 10000IU of human cho- rionic gonadotropin (hCG) to trigger ovulation. Cleavage embryos were transferred after 5 days of the hCG administration. A daily dose of 40 mg Dydrogesterone was supported in the luteal phase and continued until 10 weeks of pregnancy.

Clinical pregnancy was defined as transvaginal ultrasonography show a gestation with fetal heart activity after embryo transfer 4 to 5 weeks and clinical pregnancy rate is

Amend “were” to “was”.

There is no space before “doses”.

Add “be” before “further”.

Amend “show” to “showing”.

Amend “4” to “of 4”.

On page 7,

The strength of our study is that it is the first study, as far as it is known, to correlate serum LH with the clinical pregnancy in Ovulation induction cycle frozen-thawed em- bryo transfer patients. In the present study, the predictive value of LH were similar for age. Endometrial thicknes and the number of embryos-transferred which were recognized predictive indicator[21-23]. After controlling for the other independent variables (age, AMH, basal FSH, the number of retrieved oocytes), the significant association between serum LH and clinical pregnancy remained very strong (OR was 1.02, 95% CI: 1.0-1.03, P=0.02).

5. conclusion

In conclusion, our study demonstrates that OI-FET has a certain effect on pregnancy outcomes besides AFC, age, endometrial thickness, number of transplanted embryos, number of embryos reported in previous studies[24]. Paitients that younger than 37.5 years old, with more than 1.5 transplanted embryos and 4.5 quality embryos and LH lev- els were lower than 8.46mIU/mL on hCG trigge day have high clinical pregnancy rate and good clinical outcome. Therefore, The LH level on the hCG trigger day can be considered as a reference indicator for predicte the outcome of the FET and guide clinical decision- making.

Please amend “were” to “was”.

Amend “thicknes” to “thickness”

Remove “which”.

Add “as” before “predictive”.

Amend “indicator” to “indicators”.

In conclusion,

Amend “Paitients” to “Patients”.

Amend “trigge” to “trigger”.

Amend “predicte”. How about “predicting”?

Amend “guide” to “guiding”.

Author Response

Point 1: What percentage of patients in the non-pregnant group were lower than an LH value of 8.46 mIU/mL? If it is not o%, it is better to discuss the reason why they have not been pregnant.

Response 1:  In this study, LH was proposed as a neglected indicator, which in fact has certain guiding significance for clinical decision-making. Although the LH value is lower than 8.46 mIU/mL, there may be other influencing factors in non-pregnant patients, such as embryonic factors, which have been supplemented in the discussion section according to the reviewer's comments.

Point 2: Similar to this study, are there any reports focusing on the LH value? Are there differences in LH levels among races? Because there may be factors involved with the LH value, for example, lifestyles, such as nutrients, smoking, and alcohol.

Response 2:  There is no relevant literature in FET that focuses on the predictive value of clinical outcome of LH. There are indeed large individual differences in LH levels. This paper mainly studies Chinese women, who have similar lifestyles, such as no smoking history. Of course, we will further expand the research data and scope in the future, hoping to provide more accurate LH guidance standards for different groups.

Point 3: The LH value has a wide reference range, and there are large individual differences. What kind of approach should be taken to increase the pregnancy rate? Unfortunately, there is no difference in the birth rate even if the pregnancy rate increases.

Response 3:  the purpose of this paper is to provide more references for clinicians to help increase the pregnancy rate by summarizing clinical data. However, there are many factors affecting birth, such as fetal malformation, complications during pregnancy, special conditions during childbirth, and the medical level of hospital during childbirth. Therefore, it is not necessarily meaningless to observe LH for birth

Point 4:English language and style are minor spell check required

Response 4:  English language and style have been checked

Reviewer 3 Report

Good presentation of the results, but the introduction does not support fully the article. Please give a more extensive role of hormones and LH in endometrium and fet cycles. A comment on fresh ET should be made.

Author Response

Response 1 : According to the comments, background and relevant references have been added to the introduction

Response 2: English language and style have been checked